# Redox Regulation of Phosphatase and Tensin Homolog by Bicarbonate and Hydrogen Peroxide: Implication of Peroxymonocarbonate in Cell Signaling

**DOI:** 10.3390/antiox13040473

**Published:** 2024-04-17

**Authors:** Vu Hoang Trinh, Jin-Myung Choi, Thang Nguyen Huu, Dhiraj Kumar Sah, Hyun-Joong Yoon, Sang-Chul Park, Yu-Seok Jung, Young-Keun Ahn, Kun-Ho Lee, Seung-Rock Lee

**Affiliations:** 1Department of Biochemistry, Department of Biomedical Sciences, Chonnam National University Medical School, Gwangju 501190, Republic of Korea; trinhhoangvu@jnu.ac.kr (V.H.T.); 206847@jnu.ac.kr (T.N.H.); 197784@chonnam.edu (D.K.S.); hjms0320@jnu.ac.kr (H.-J.Y.); 2Department of Oncology, Department of Medical Sciences, Pham Ngoc Thach University of Medicine, Ho Chi Minh City 700000, Vietnam; 3Luxanima Inc., Room 102, 12-55, Sandan-gil, Hwasun-eup, Hwasun-gun 58128, Republic of Korea; choijm2@jnu.ac.kr; 4The Future Life & Society Research Center, Advanced Institute of Aging Science, Chonnam National University, Gwangju 61469, Republic of Korea; parksc@snu.ac.kr; 5Chonnam National University Medical School, Gwangju 501190, Republic of Korea; william67@jnu.ac.kr; 6Department of Cardiology, Chonnam National University Hospital, Gwangju 61469, Republic of Korea; cecilyk@jnu.ac.kr; 7Department of Biomedical Science, Chosun University, Gwangju 61452, Republic of Korea; leekho@chosun.ac.kr; 8Department of Neural Development and Disease, Korea Brain Research Institute, Daegu 41062, Republic of Korea

**Keywords:** PTEN redox regulation, cell signaling, H_2_O_2_, bicarbonate, peroxymonocarbonate, carbonic anhydrase

## Abstract

Phosphatase and tensin homolog (PTEN) is a negative regulator of the phosphoinositide 3-kinases/protein kinase B (PI3K/AKT) signaling pathway. Notably, its active site contains a cysteine residue that is susceptible to oxidation by hydrogen peroxide (H_2_O_2_). This oxidation inhibits the phosphatase function of PTEN, critically contributing to the activation of the PI3K/AKT pathway. Upon the stimulation of cell surface receptors, the activity of NADPH oxidase (NOX) generates a transient amount of H_2_O_2_, serving as a mediator in this pathway by oxidizing PTEN. The mechanism underlying this oxidation, occurring despite the presence of highly efficient and abundant cellular oxidant-protecting and reducing systems, continues to pose a perplexing conundrum. Here, we demonstrate that the presence of bicarbonate (HCO_3_^−^) promoted the rate of H_2_O_2_-mediated PTEN oxidation, probably through the formation of peroxymonocarbonate (HCO_4_^−^), and consequently potentiated the phosphorylation of AKT. Acetazolamide (ATZ), a carbonic anhydrase (CA) inhibitor, was shown to diminish the oxidation of PTEN. Thus, CA can also be considered as a modulator in this context. In essence, our findings consolidate the crucial role of HCO_3_^−^ in the redox regulation of PTEN by H_2_O_2_, leading to the presumption that HCO_4_^−^ is a signaling molecule during cellular physiological processes.

## 1. Introduction

When cells are stimulated by growth factors and cytokines, such as platelet-derived growth factor (PDGF), epidermal growth factor (EGF), insulin, granulocyte–macrophage colony-stimulating factor (GM-CSF), tumor necrosis factor-α (TNF-α), interleukin-1 (IL-1), and interleukin-3 (IL-3), H_2_O_2_ is produced through the activity of NOXs [1]. The activation of NOXs results in the production of superoxide (O_2_^•−^), which is subsequently transformed to H_2_O_2_ by the activity of superoxide dismutase (SOD) [2]. This physiological H_2_O_2_ influences various intracellular signaling pathways. Its capacity to oxidize the cysteine residues of some proteins, such as protein tyrosine phosphatases (PTPs), results in a functional modification that notably impairs their activities [3]. PTPs play pivotal roles in cellular processes involving cell growth, proliferation, and differentiation [4]. Within their structural framework, all PTPs feature a cysteine residue in their active site [5]. When exposed to H_2_O_2_, the cysteine residue undergoes oxidation, transforming into cysteine–sulfenic acid (Cys-SOH) and leading to the oxidative inhibition of PTP’s enzymatic activity [6].

PTEN, a member of the PTP family, structurally comprises five functional domains: a short N-terminal phosphatidylinositol (PtdIns)(4,5)P2-binding domain (PBD); a catalytic phosphatase domain; a C2 lipid/membrane-binding domain; a C-terminal tail containing Pro, Glu, Ser, and Thr (PEST) sequences; and a class I PDZ-binding (PDZ-BD) motif. PTEN interacts with phospholipid membranes through the C2 domain located at the C-terminal end. The tight linkage between the C2 and phosphatase domains implies that the C2 domain not only aids in recruiting PTEN to the membrane but also optimizes the orientation of the catalytic domain associated with its membrane-bound substrate. Moreover, the phosphatase domain of PTEN has an enlarged active site to accommodate its phosphoinositide substrate, membrane-bound phosphatidylinositol-3,4,5-triphosphate (PIP3) [7,8]. Hence, PTEN is considered a tumor suppressor owing to its ability to dephosphorylate PIP3 and, consequently, negatively regulate the PI3K/AKT signaling pathway, which is pivotal in controlling cell survival, growth, and proliferation [9,10]. In addition to cancer studies, substantial insights into PTEN’s role in physiological processes are available. Notably, PTEN inhibition has been demonstrated as a promising therapeutic intervention for neurodegenerative diseases, ischemia, infection, and insulin-resistant metabolic disorders [11,12,13].

Similar to other members of the PTP family, PTEN contains a cysteine residue in the active site of its phosphatase domain, rendering it susceptible to oxidative inhibition by reactive oxygen species (ROS), particularly H_2_O_2_ [5]. Lee et al. were the first to demonstrate the reversible inactivation of PTEN by H_2_O_2_ through the oxidation of the Cys124 catalytic residue in the active site and the formation of an intramolecular disulfide bond with Cys71 residue. This inactivation is reversible because oxidized PTEN can be converted back to the functional reduced form by cellular reducing agents, such as the Thioredoxin/Thioredoxin Reductase (Trx/TrxR) system [14]. Additionally, in the cell, Peroxiredoxins (Prx), thiol-specific antioxidants of the peroxidase family, can function as modulators of H_2_O_2_-induced phosphorylation signaling due to their capacity to sense and eliminate peroxides [15,16,17]. Thus, the mechanism of PTEN oxidation by transient H_2_O_2_, a signaling molecule produced in response to growth factor receptor stimulation, raises questions about how this small amount of physiological H_2_O_2_ can function to oxidize PTEN in a cellular environment rich in H_2_O_2_ scavengers such as catalase, glutathione peroxidase, and Prx, alongside the presence of ubiquitous Trx/TrxR reducing systems.

It has been known since the 1980s that H_2_O_2_ can react with bicarbonate/carbon dioxide (HCO_3_^−^/CO_2_) to form HCO_4_^−^: H_2_O_2_ + HCO_3_^−^/CO_2_ ⇌ HCO_4_^−^ + H_2_O/H^+^ [18,19]. In a neutral-pH aqueous solution, the HCO_4_^−^ formation process occurs rapidly at 25 °C with a half-life t_1/2_ of 10 min or below [20,21]. CAs and a zinc model complex can accelerate this reaction [19,21]. HCO_4_^−^ is a robust oxidant with a higher catalytic potency comparable to that of H_2_O_2_. Its interaction with target molecules proceeds at velocities ranging from 100 to 1000 times faster than those observed with H_2_O_2_ [22]. Sulfide oxidation caused by HCO_4_^−^ transcends that caused by H_2_O_2_ by approximately 300-fold [20]. Notably, HCO_4_^−^ also serves as a potent two-electron oxidant primarily accountable for biothiol peroxidation [23]. Hence, the presence of HCO_3_^−^ is demonstrated to be a pivotal factor in promoting the oxidative reactivity of H_2_O_2_ towards PTPs, such as PTP1B, SHP-2, and PTPN22, during signal transduction processes [24,25,26]. When cells are stimulated by growth factors, HCO_3_^−^ is produced through the activation of a transmembrane enzyme, CA IX, which catalyzes the hydration of CO_2_ (CO_2_ + H_2_O → HCO_3_^−^ + H^+^) [27]. Dagnell et al. demonstrated that HCO_3_^−^ not only facilitates but also plays an essential role in the H_2_O_2_-mediated inactivation of PTP1B, surpassing the protection provided by the Trx/TrxR reducing systems [25]. Given that PTEN belongs to the PTP family, it is worth exploring whether HCO_3_^−^ may somehow affect the H_2_O_2_-mediated oxidative inactivation of PTEN. Based on the available pieces of information, we anticipated that the presence of HCO_3_^−^ in combination with H_2_O_2_, through the formation of HCO_4_^−^, would augment the oxidation of PTEN.

## 2. Materials and Methods

### 2.1. Materials

Dulbecco’s modified Eagle’s medium (DMEM) D5648 and N-Ethylmaleimide (NEM) were sourced from Sigma-Aldrich (St. Louis, MO, USA), 1 M HEPES from Enzynomics (Daejeon, Republic of Korea), 100X penicillin/streptomycin from Capricorn Scientific (Ebsdorfergrund, Germany), fetal bovine serum (FBS) from Welgene (Gyeongsan, Republic of Korea), 3% hydrogen peroxide from Samchun (Seoul, Republic of Korea), sodium bicarbonate from Amresco (Solon, OH, USA), Pro-prep Protein Extraction Solution from iNtRON Biotechnology (Seongnam, Republic of Korea), a PageRuler Prestained Protein Ladder No. 26616 and BCA Protein Assay Kit from Thermo Scientific (Waltham, MA, USA), and ATZ from MedChemExpress (New York City, NJ, USA). The antibodies used for Western blotting comprised primary PTEN antibody, β-actin, phosphorylated-Serine-473 AKT, total AKT and anti-rabbit immunoglobulin G horseradish peroxidase-linked secondary antibodies. All other chemicals and reagents were of analytical grade.

### 2.2. Cell Culture 

HepG2 cells, originally obtained from the American Type Culture Collection (Manassas, VA, USA), were cultured in regular DMEM supplemented with 10% FBS and 1X penicillin/streptomycin.

### 2.3. Immunoblot Analysis of H_2_O_2_-Induced Oxidation of PTEN 

To assess the redox state of PTEN, we performed mobility shift assays in cells, employing N-Ethylmaleimide (NEM) as an alkylating agent, as described in our previous publication [28]. HepG2 cells were cultured until they reached 90% confluency. Subsequently, they were washed with PBS and switched to media containing 0.1% FBS and 25 mM HEPES, with various sodium bicarbonate or ATZ conditions, and then pre-incubated at 37 °C with 0.1% CO_2_. Similar serum-free stimulation media containing 0.5–1 mM H_2_O_2_ were prepared before being applied to the cells. After pre-incubation for 4 h, the media were removed, and cells were incubated with the prepared stimulation media. Subsequently, the stimulation media were removed at varying time points, and the cells were washed twice using cold PBS. The reaction was stopped by adding Pro-prep lysis buffer containing 10 mM NEM. Cell lysates were collected, and the protein concentrations were quantified using the BCA method. Subsequently, the samples were subjected to non-reducing SDS-PAGE, followed by Western blotting using PTEN, β-Actin, pAKT, and total AKT-specific antibodies (Figure 1).

### 2.4. Statistical Analysis 

PTEN oxidation levels in the samples were quantified by analyzing the oxidized/reduced PTEN states using ImageJ version 1.54d. Data significances were evaluated using Student’s *t* test. Differences were significant when *p* < 0.05. Graphical values are presented as the mean ± standard error (SE). The trendlines for PTEN oxidation and AKT phosphorylation represent linear regression. 

## 3. Results

### 3.1. The Presence of HCO_3_^−^ Potentiates the PTEN Oxidation by H_2_O_2_

HepG2 cells were pre-incubated in HCO_3_^−^-free media at 37 °C with 0.1% CO_2_ for 4 h. Various stimulation media were made before being administered to the cells: 44 mM HCO_3_^−^ alone, 1 mM H_2_O_2_ alone, a combination of 1 mM H_2_O_2_ and 22 mM HCO_3_^−^, and a combination of 1 mM H_2_O_2_ and 44 mM HCO_3_^−^. The percentages of oxidized PTEN were assessed after 10 min of treatment. The results showed that with the presence of 22 mM or 44 mM HCO_3_^−^ in the stimulation media, the oxidations of PTEN in 10 min were significantly higher (*p* < 0.05) compared to the cells which were treated with H_2_O_2_ alone. The oxidation did not exhibit an increased magnitude when more HCO_3_^−^ was added. In addition, stimulation media with only 44 mM HCO_3_^−^ exhibited no oxidation effect (Figure 2).

### 3.2. The Presence of HCO_3_^−^ Facilitates Redox Regulation of PTEN by H_2_O_2_

To investigate the change in PTEN oxidation, we observed the percentages of oxidized PTEN in different time courses. Treatments of HepG2 cells by H_2_O_2_, with the presence and absence of 44 mM HCO_3_^−^, in 5, 10, 15, 30 60, and 120 min, resulted in PTEN oxidation in various time-dependent manners. In the first group, cells were pre-incubated and treated in stimulation media containing 44 mM HCO_3_^−^. In the second group, the condition was similar except that the media was HCO_3_^−^-free. Our data indicated that in the HCO_3_^−^-containing group, the PTEN oxidation rate soared to its peak at 10 min and then decreased gradually. At 60 min, almost all oxidized PTENs were reduced back, and the recovery was completed by 120 min. In the HCO_3_^−^-free group, the oxidation rate increased at a slower speed, reaching its peak at 30 min. At 60 min, there was a considerable amount of oxidized PTEN that was not reduced back, and the amount was much higher than that observed in the HCO_3_^−^-containing group. In addition, the maximum oxidation level was higher in the HCO_3_^−^-containing group. The linear regression trendlines illustrate that PTEN oxidation was faster in the HCO_3_^−^-containing group than in the HCO_3_^−^-free group. Comparing the slopes between the trendlines (2.87 and 1.16), the difference in oxidation speed was nearly 2.5 times. During the recovery period, the absence of HCO_3_^−^ remarkably impaired the reduction of oxidized PTEN. A comparison of the slopes revealed that the difference in reduction speed was also approximately equivalent (2.3-fold). These results clearly demonstrate that HCO_3_^−^ accelerates the H_2_O_2_-mediated redox regulation of PTEN (Figure 3).

### 3.3. The Activation of PI3K/AKT Pathway via PTEN Oxidation by H_2_O_2_ and HCO_3_^−^

PTEN oxidation inhibits its phosphatase function and subsequently leads to the activation of the PI3K/AKT signaling pathway [10]. We have shown that PTEN can be oxidized at a faster speed by the combination of HCO_3_^−^ and H_2_O_2_ than by H_2_O_2_ alone. To investigate how this PTEN oxidative inhibition affects the phosphorylation of AKT, we conducted the same protocol; cell lysates were subjected to SDS-PAGE with antibodies specific for pAKT and total AKT. With the addition of 44 mM HCO_3_^−^, AKT phosphorylation started to rise significantly following 10 min of exposure to H_2_O_2_, reaching the maximum value at 15 min before gradually declining. Meanwhile, the absence of HCO_3_^−^ delayed this process, with a slower increase to its peak at 30 min. Furthermore, the HCO_3_^−^-containing group also exhibited a higher proportion of pAKT across all timepoints. Comparing the slopes between the trendlines (0.0704 and 0.0263), the disparity in phosphorylation velocity was approximately 2.7-fold. Overall, under stimulation by H_2_O_2_, the presence of HCO_3_^−^ potentiates the phosphorylation of AKT in a time-dependent manner (Figure 4).

### 3.4. Oxidation of PTEN Is Decreased in the Presence of CA Inhibitor ATZ

CAs catalyze the following reaction: CO_2_ + H_2_O ⇌ HCO_3_^−^ + H^+^ [29]. We utilized a CA inhibitor, ATZ, to further investigate PTEN oxidation by H_2_O_2_ in condition with 0.1% CO_2_ and lacking HCO_3_^−^. Cells were pre-incubated for 4 h with 0.1% CO_2_ in HCO_3_^−^-free media and supplemented with various concentrations of ATZ such as 0, 0.01, 0.1, 1, and 10 µM. After the pre-incubation period, cells were exposed to 0.5 mM H_2_O_2_ in stimulation media under the same condition for 10 min. The results showed 10 µM of ATZ alone did not cause any PTEN oxidation. The presence of 1 or 10 µM of ATZ in combination with 0.5 mM H_2_O_2_ was significantly effective compared to H_2_O_2_ alone (*p* < 0.05). As the ATZ concentration increased, the PTEN oxidation decreased in a dose-dependent manner. At 10 µM, the percentage of oxidized PTEN was considerably lower than when ATZ was absent (Figure 5).

## 4. Discussion

An elevated PI3K/AKT signaling pathway has been shown to be advantageous in specific physiological processes relating to tissue regeneration; thus, inhibiting PTEN, a suppressor of this pathway, has emerged as a potential therapeutic approach for ischemia, neurodegenerative diseases, infection, and insulin-resistant metabolic disorders [11,12,13]. In addition, the activation of receptor tyrosine kinases (RTKs) is a pivotal event in transmitting phosphorylation signals upon stimulation by growth factors, therefore holding substantial importance in cell physiology [30]. In this situation, a transient amount of H_2_O_2_ is generated by membrane NOXs [31]. Subsequently, H_2_O_2_ causes the reversible oxidative inhibition of PTEN, leading to an increase in PI3K/AKT downstream cascades and, consequently, eliciting cellular effects [32,33]. The oxidative inactivation of PTEN by ROS has also been demonstrated to play a significant positive role in those physiological processes demanding cell growth [34]. It can promote cardiac remodeling [35], neuronal regeneration [36], immune response [37], glucose metabolism [38], and myogenesis [39]. However, within the cellular environment, H_2_O_2_ can be eliminated by thiol proteins from the Prx family. Furthermore, oxidized PTEN and Prx can be rapidly converted back to their active reduced forms via the Trx/TrxR/NADPH system [40,41]. Thus, PTEN phosphatase catalytic activity is conserved by Trx/TrxR and Prx systems [16,17,40,42]. Yet, in this context, PTEN is inhibited during PI3K/AKT signaling pathway activation. Additionally, under RTK stimulation, CA IX undergoes phosphorylation and engages in catalyzing the conversion of CO_2_ to HCO_3_^−^ [27]. Through our experiment results, we presume the factor that facilitates PTEN oxidation by H_2_O_2_ in the presence of HCO_3_^−^ is the formation of HCO_4_^−^ (Figure 6).

The addition of 44 mM sodium bicarbonate to DMEM D5648, which lacks HCO_3_^−^, replicates the common condition of culture media. Sodium bicarbonate can alter the pH of the tumor microenvironment, which has been demonstrated to critically affect cancer cell apoptosis and survival [43]. Therefore, we supplemented the media with HEPES as a pH buffer. Previous investigations on NIH 3T3 cells exposed to H_2_O_2_ in standard DMEM revealed reversible PTEN oxidation in a time-dependent manner. Incubation with 0.5 mM H_2_O_2_ resulted in increased levels of oxidized PTEN, peaking at 10 min before declining gradually, suggesting that PTEN was initially inactivated by H_2_O_2_ and then reactivated by cellular reductants as the H_2_O_2_ declined. This reversible PTEN inactivation was also observed in HeLa cells exposed to H_2_O_2_ [14]. In our recent studies, cells were treated with H_2_O_2_ for specific durations, extending the time points to 120 min to ensure the completion of the reduction process. Similarly, oxidized PTEN exhibited a time-dependent increase followed by a decline. The PTEN oxidation rate reached its peak after exposure for 10 min, and the oxidized PTEN was completely converted to the reduced form within 120 min of exposure. Based on these results, it is evident that treatments with H_2_O_2_ in standard HCO_3_^−^-containing media yield maximum PTEN oxidation after 10 min of exposure, followed by the conversion of oxidized PTEN to its reduced state by the cellular Trx/TrxR system within approximately 60 min after exposure [28,42]. Therefore, we initially tested the role of HCO_3_^−^ in PTEN oxidation by H_2_O_2_ through evaluating the proportions of oxidized PTEN following 10 min of exposure to stimulation media. The pre-incubation period aimed to diminish the HCO_3_^−^ in both the extracellular and intracellular environments. Our findings indicated HCO_3_^−^ alone did not cause any PTEN oxidation. Meanwhile, H_2_O_2_ alone had a significantly lower percentage than the H_2_O_2_-HCO_3_^−^ combination. Additionally, there was no notable difference between the HCO_3_^−^ concentration of 22 mM and 44 mM. This implies that the greater percentages observed in the H_2_O_2_-HCO_3_^−^ combination groups were not merely the cumulative effect of both H_2_O_2_ and HCO_3_^−^_._ A plausible explanation is that HCO_3_^−^ can empower the catalytic activity of H_2_O_2_, supporting the hypothesis that there might be the formation of HCO_4_^−^, a more reactive oxidant.

Given that HCO_4_^−^ was demonstrated to exhibit a greater oxidation rate towards sulfide when compared to H_2_O_2_ [20], we suppose that it may also influence PTEN oxidation. This implies that the peak of PTEN oxidation in each group may not be solely at the 10 min timepoint. Therefore, we extended experiments to additional time courses to investigate the redox regulation of HCO_3_^−^ and H_2_O_2_. Additionally, we reduced the concentration of H_2_O_2_ to 0.5 mM and anticipated it could make the difference more visible. A similar pattern to results of our previous studies was observed this time on HepG2 cells in 44 mM HCO_3_^−^-supplemented media. PTEN was predominantly oxidized within 5–15 min, with the highest oxidation rate being observed at 10 min, then reduced back to the active form. Thus, the PTEN oxidation observed in the HCO_3_^−^-containing group in the present study follows a trend consistent with our former studies.

The exclusion of HCO_3_^−^ from the experimental conditions considerably prolonged the PTEN oxidation period, requiring up to 30 min to reach its peak. However, the H_2_O_2_-mediated oxidative inactivation of intracellular PTPs in signaling events typically happens swiftly within the timeframe of 5–15 min, typically peaking at 10 min [14,44,45]. This disparity suggests that the absence of HCO_3_^−^ adversely affects the PTEN oxidation speed. Our result aligns with the findings of Dagnell et al., indicating that HCO_3_^−^ can potentiate the H_2_O_2_-dependent oxidative inactivation of PTP [25]. Illustrative data analysis suggests that the combination of HCO_3_^−^ and H_2_O_2_ accelerated the rate of PTEN oxidation in the cellular environment by approximately 2.5-fold in a time-dependent manner. Notably, this was also strongly correlated to the similar decrease in the PTEN reduction speed in the HCO_3_^−^-free group. Based on growing evidence about the role of HCO_3_^−^ in H_2_O_2_-mediated oxidation under cell signaling processes, we suppose the formed HCO_4_^−^ promotes PTEN redox regulation and therefore exerts function as a signaling molecule.

PTEN acts as a negative regulator of the PI3K/AKT signaling pathway, which induces various cellular processes including protein synthesis, cell survival, proliferation, and migration. This mechanism plays an important role in human physiological and pathological conditions [7,12,46]. Following the result that H_2_O_2_-mediated oxidative inhibition of PTEN is potentiated and accelerated by HCO_3_^−^, we examined its subsequent impact on the phosphorylation of AKT. The elevating trends of pAKT/AKT reflect similar patterns observed in PTEN oxidation, both in HCO_3_^−^-containing and HCO_3_^−^-free status. Our findings are consistent with previous evidence suggesting that H_2_O_2_ can activate the PI3K/AKT pathway by oxidizing PTEN [10]. Additionally, the time-dependent increase in pAKT/AKT is equivalent to the rise in PTEN oxidation in the presence of HCO_3_^−^, which is 2.7-fold and 2.5-fold, correspondingly. In another words, without HCO_3_^−^, PTEN oxidation is impaired, leading to the delay in AKT phosphorylation. Thus, in this context, the increase in AKT activation arises from the surge in PTEN oxidation. Considering HCO_4_^−^ is formed through the reaction between H_2_O_2_ and HCO_3_^−^, we propose that HCO_4_^−^ can positively regulate the PI3K/AKT signaling pathway via the oxidative inactivation of PTEN.

However, we also concede a possibility that a minor amount of HCO_3_^−^ might be produced in the media through the reaction of CO_2_ and H_2_O, despite the CO_2_ level being minimized to 0.1% in our experiments. The CO_2_ hydration reaction to form HCO_3_^−^ is reversible and catalyzed by CAs [29]. Notably, CA IX, a transmembrane isoform, is often over-expressed in various types of cancer cells as it is associated with hypoxia-inducible factor 1 (HIF1) [47]. Thus, we had expected utilizing ATZ, a CA inhibitor, to prevent the production of HCO_3_^−^. Our prior data indicated that PTEN oxidation by H_2_O_2_ in HCO_3_^−^-free media was significantly lower than that in HCO_3_^−^-containing media. However, when 0.01–10 mM ATZ was added, the oxidation decreased further in a dose-dependent manner. Given that no oxidized PTEN bands were detected in the presence of 10 mM ATZ alone, we acknowledged that ATZ did not directly cause PTEN oxidation in these conditions. A possible explanation could be that a higher dose of ATZ more effectively impedes the conversion of CO_2_ to HCO_3_^−^. Consequently, the formation of HCO_4_^−^ from HCO_3_^−^ and H_2_O_2_ was also reduced. This supports our presumption that HCO_4_^−^ plays a critical role in PTEN oxidation under stimulation by H_2_O_2_. 

A limitation of our study is the lack of a direct evaluation regarding the actual concentration of formed HCO_4_^−^ when combining H_2_O_2_ and HCO_3_^−^. We could only adjust indirectly through conducting different conditions of CO_2_, HCO_3_^−^, and CA inhibitor. 

In summary, the results of our experiment support the hypothesis that the presence of HCO_3_^−^ promotes the H_2_O_2_-mediated inactivation of PTEN. This subsequently promotes the activation of the PI3K/AKT signaling pathway. In addition, the reduction of oxidized PTEN by antioxidant systems is also equivalently affected. A plausible explanation could be the formation of HCO_4_^−^. CAs are prospective targets for modulating this process as they are involved in HCO_3_^−^ production. In the future, further experiments would be conducted to consolidate the role of HCO_4_^−^ in the redox regulation of PTEN.

## Figures and Tables

**Figure 1 antioxidants-13-00473-f001:**
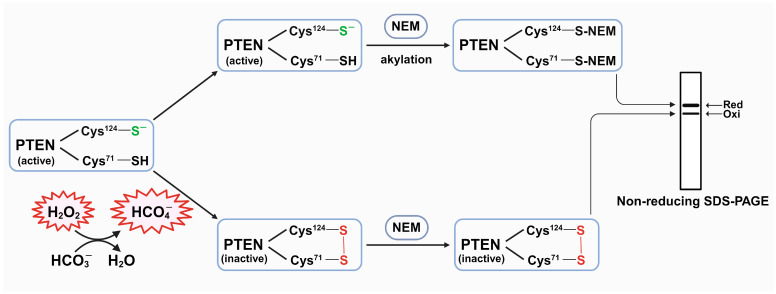
Experimental scheme for detecting redox status of PTEN by mobility shift. Upon HCO_4_^−^ exposure, a proportion of PTEN can be oxidized in the Cystein^124^ residue of their active site through creating a reversible intracellular disulfide bond with Cystein^71^. Subsequently, the supplement of NEM akylates reduced PTEN to a PTEN-NEM complex, increasing the molecular weight, thereby resulting in a lower shift in non-reducing SDS-PAGE.

**Figure 2 antioxidants-13-00473-f002:**
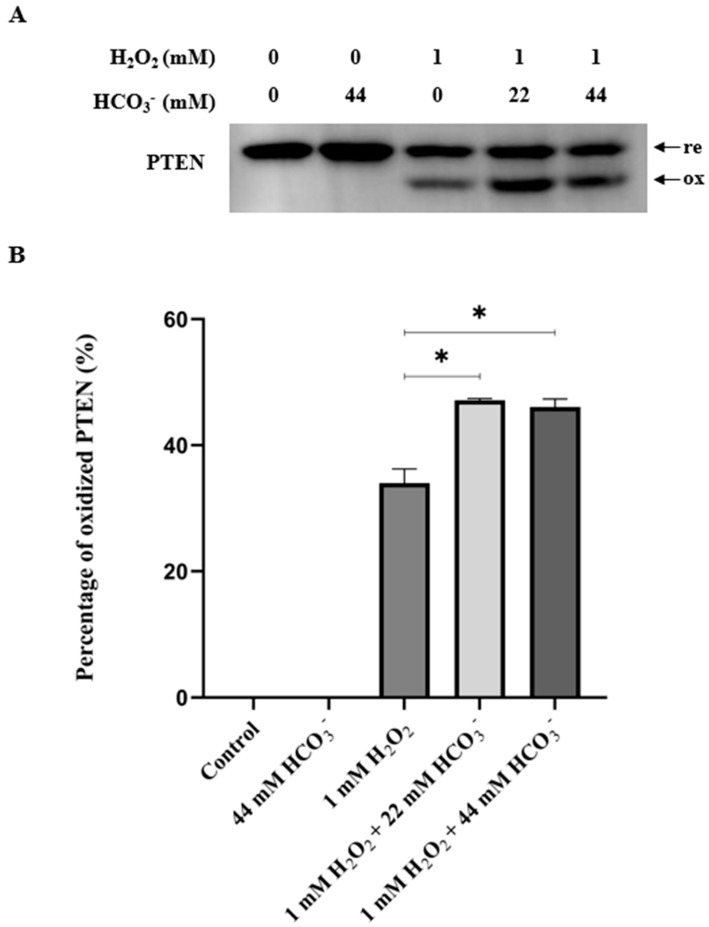
Effect of HCO_3_^−^ on oxidation of PTEN by H_2_O_2_. HepG2 cells were cultured until they reached 90% confluency, then washed with PBS and transferred to DMEM (L-glutamine, 1X penicillin/streptomycin, 0.1% FBS, and 25 mM HEPES) without 44 mM sodium HCO_3_^−^, and the pH was neutral. Then, the cells were incubated at 37 °C with 0.1% CO_2_ for 4 h and, subsequently, treated with prepared stimulation media containing 44 mM HCO_3_^−^ alone, 1 mM H_2_O_2_ alone, a combination of 1 mM H_2_O_2_ and 22 mM HCO_3_^−^, or a combination of 1 mM H_2_O_2_ and 44 mM HCO_3_^−^. After 10 min, the reaction was stopped by adding lysis buffer containing protease inhibitors and 10 mM NEM. Then, the cell lysates were collected and used for SDS-PAGE and Western blotting analyses. The results obtained using PTEN antibody are presented in (**A**). The quantification result (**B**) indicated that with the presence of 22 mM or 44 mM HCO_3_^−^, PTEN oxidation was significantly higher than with treatment with only H_2_O_2_ (* *p* < 0.05). The 44 mM HCO_3_^−^ alone scenario showed no oxidation effect.

**Figure 3 antioxidants-13-00473-f003:**
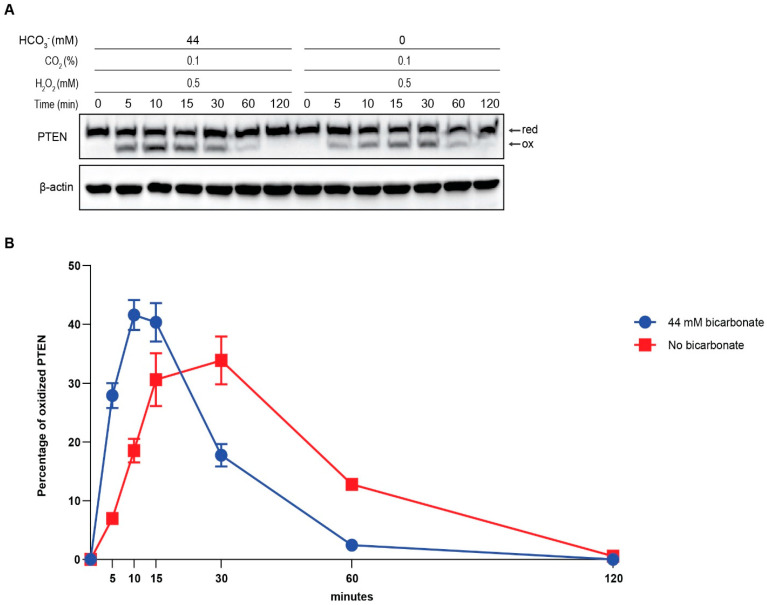
HCO_3_^−^ facilitates the redox regulation of PTEN by H_2_O_2_. HepG2 cells were cultured until they reached 90% confluency, then washed with PBS and transferred to DMEM (L-glutamine, 1X penicillin/streptomycin, 0.1% FBS, and 25 mM HEPES) with or without 44 mM sodium bicarbonate, and the pH was neutral. Then, the cells were incubated at 37 °C with 0.1% CO_2_ for 4 h and, subsequently, treated with prepared stimulation media containing 0.5 mM H_2_O_2_. After 5, 10, 15, 30, 60, and 120 min, the reaction was stopped by adding lysis buffer containing protease inhibitors and 10 mM NEM. Then, the cell lysates were collected and used for SDS-PAGE and Western blotting analyses. The results obtained using PTEN and β-actin antibodies are presented in (**A**). The quantification results (**B**) indicate that in the absence of HCO_3_-, the rate of PTEN oxidation diminished, and the reduction period was prolonged. Data are presented as the mean ± SE of three independent experiments. The linear regression trendlines of PTEN oxidation and reduction rates presented in (**C**) show that the presence of HCO_3_- accelerates the H_2_O_2_-mediated redox regulation of PTEN.

**Figure 4 antioxidants-13-00473-f004:**
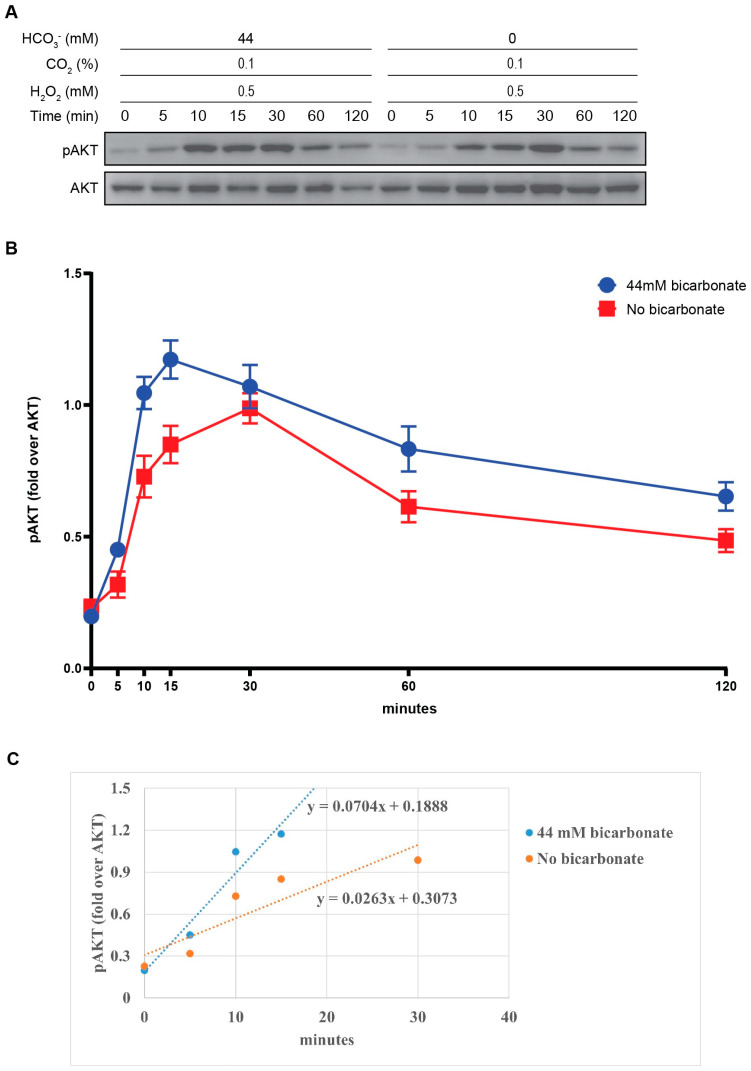
HCO_3_^−^ accelerates and potentiates the phosphorylation of AKT induced by H_2_O_2_. HepG2 cells were pre-incubated and exposed to 0.5 mM H_2_O_2_ in HCO_3_^−^-supplemented or HCO_3_^−^-free media. After 5, 10, 15, 30, 60, and 120 min, cell lysates were collected and subjected to Western blot using pAKT and total AKT antibody (**A**). Data are quantified by the pAKT fold over total AKT and presented as mean ± SE. The combination of HCO_3_^−^ and H_2_O_2_ resulted in faster and higher AKT phosphorylation than H_2_O_2_ alone (**B**). The linear regression trendlines of pAKT/AKT presented in (**C**) show that the presence of HCO_3_^−^ elevated the AKT phosphorylation rate.

**Figure 5 antioxidants-13-00473-f005:**
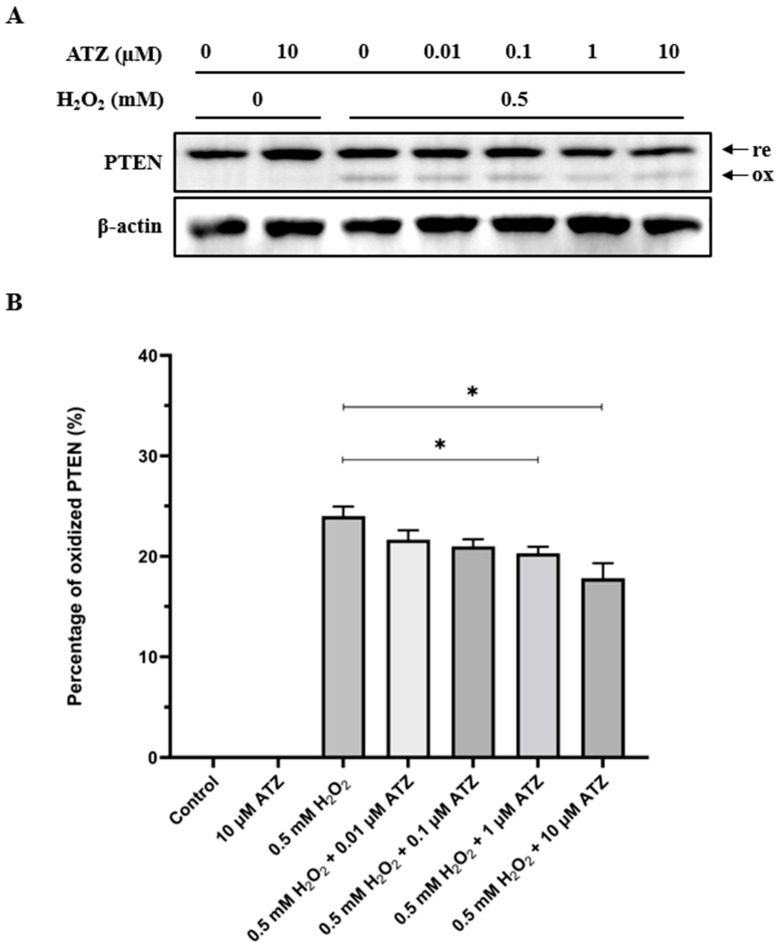
Effect of ATZ, a CA inhibitor, on oxidation of PTEN by H_2_O_2_. HepG2 cells were pre-incubated in HCO_3_^−^-free media, supplemented with 0, 0.01, 0.1, 1, and 10 µM ATZ, at 37 °C with 0.1% CO_2_ for 4 h. Then, they were changed to stimulation media with same components plus 0.5 mM H_2_O_2_. After 10 min, the reaction was terminated by NEM-containing lysis buffer. Western blot data with PTEN and β-actin antibodies are shown (**A**). ATZ considerably impeded PTEN oxidation in a dose-dependent manner (**B**). Stars are considered as significances (*p* < 0.05).

**Figure 6 antioxidants-13-00473-f006:**
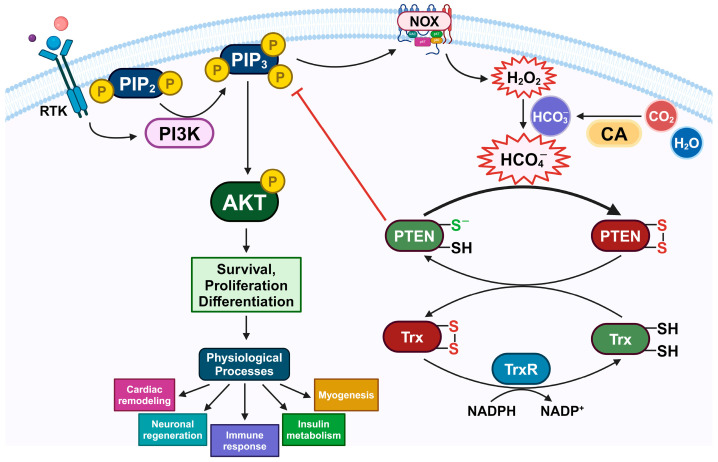
Schematic role of HCO_4_^−^ in the redox regulation of PTEN and cell signaling. The stimulation of RTK induces the assembly of PI3K to phosphorylate PIP2 to PIP3, subsequently activating the AKT signaling cascade. PTEN can negatively modulate this pathway. During this condition, H_2_O_2_, produced through the activity of NOXs, oxidizes PTEN. In the cellular environment, PTEN oxidation can be impaired and reversed by Trx/TrxR/NADPH reducing systems. HCO_3_^−^ can also be generated through the activity of CAs. In the presence of HCO_3_^−^ together with H_2_O_2_, a higher reactive oxidant HCO_4_^−^ is formed. Hence, HCO_4_^−^ is supposed to accelerate the oxidative inactivation of PTEN, consequently increase the PI3K/AKT signaling pathway, and, as a result, promote cell survival, proliferation, and differentiation in physiological processes.

## Data Availability

The data presented in this study are available on request from the corresponding author.

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
