# Peer review of "Redox Regulation of Phosphatase and Tensin Homolog by Bicarbonate and Hydrogen Peroxide: Implication of Peroxymonocarbonate in Cell Signaling"

_antioxidants, 2024, doi:10.3390/antiox13040473_

Round 1

Reviewer 1 Report

Comments and Suggestions for Authors

Line 79-81: "It has been known since 1980s that H2O2 can react with bicarbonate/carbon dioxide (HCO3-/CO2) to form peroxymonocarbonate (HCO4-)-: H2O2 + HCO3-/CO2 ⇌ _HCO4- + H2O/H+." This sentence seems important since it appears to be why the authors worked on this paper. Please cite references.

Line 81-82: "In neutral pH aqueous solution, HCO4- formation process occurs rapidly at 25 with half-life t1/2 of 10 min or below." Please cite references.

The authors should explain why they selected 44 mM bicarbonate. Can a 44 mM bicarbonate condition be obtained for normal and cancer cells?

Please refer doi: 10.1007/s11427-016-0373-3. A higher concentration of bicarbonate can be used in cancer treatment. It is well known that the pH of cancer cells is usually acidic.

Figure 5 is not acceptable due to the fact that the condition doesn't exist in normal and cancer cells.

Comments on the Quality of English Language

I feel that Figure 5 is a too exaggerated figure.

Reviewer 2 Report

Comments and Suggestions for Authors

Authors described Redox Regulation of PTEN by Bicarbonate and Hydrogen Peroxide mediated by Peroxymonocarbonate. This is a highly innovative study of novel redox mediator peroxymonocarbonate in cell signaling, particularly regulation of phosphatase PTEN. The critical role of Cysteine residues in PTEN function is known for many years, however, specific redox molecular mechanism of PTEN redox reactions are still elusive. In this manuscript, authors directly tested PTEN redox reactions with peroxymonocarbonate in HepG2 cells.  They demonstrated modulation of downstream AKT phosphorylation in response to peroxymonocarbonate produced by reaction of H2O2 and HCO3-(CO2). This is a very interesting work.

There are a couple of minor issues related to limitations of experimental approach which authors should comment in the discussion section.

1) Authors incubated cells with very high H2O2 levels (0.5 mM) for up to 120 minutes. This may affect the cell viability. Authors showed time 0 min as well as H2O2 without bicarbonate, however, high levels of H2O2 may have alter cell function. Please also comment if authors tried smaller H2O2 concentrations.

2) Figure 4A,B shows data with CO2+bicarbonate vs CO2 only. It is possible that doing incubation without CO2 would increase the difference since CO2 produces small bicarbonate levels in the media. Please comment the potential contamination of CO2 only with bicarbonate both in Figure 3 and 4 studies.

Round 2

Reviewer 1 Report

Comments and Suggestions for Authors

The manuscript is well written.

Comments on the Quality of English Language

English is better to recheck.